# HBD Inhibits the Development of Colitis-Associated Cancer in Mice via the IL-6R/STAT3 Signaling Pathway

**DOI:** 10.3390/ijms20051069

**Published:** 2019-03-01

**Authors:** Song Deng, Aiping Wang, Xi Chen, Qun Du, Yanli Wu, Gang Chen, Wenfeng Guo, Yanwu Li

**Affiliations:** 1Pi-Wei Institute, Guangzhou University of Chinese Medicine, Guangzhou 510405, China; ds_xmy_001@163.com (S.D.); 15817146870@163.com (A.W.); 18520729545@163.com (X.C.); duqun@gzucm.edu.cn (Q.D.); littlecrystals@126.com (Y.W.); 2School of Traditional Chinese Materia Medica, Shenyang Pharmaceutical University, Shenyang 110016, China; chengang1152001@163.com

**Keywords:** colitis-associated cancer, HBD, IL-6Rα/STAT3

## Abstract

Colitis-associated cancer (CAC) is a malignant disease of the colon that is caused by recurrent episodes of chronic intestinal inflammation. Huangqi Baizhu decoction (HBD) is a classic prescription comprised of *Radix Astragali* and *Rhizoma Atractylodis*, which are usually used to treat digestive conditions, such as peptic ulcers, colitis, or colorectal carcinoma in clinics. HBD is well known for “tonifying qi and spleen” based on the theories of traditional Chinese medicine, and has the preponderant effect of alleviating chronic intestinal mucosa damage associated with disease. However, the underlying mechanism behind this is still unknown. In the current study, we employed the AOM/DSS mouse model to analyze the effects of HBD on the development of inflammation in colonic carcinoma. The in vivo study showed that HBD could significantly reduce the mortality of mice and control the incidence and size of colonic tumors by inhibiting the IL-6/STAT3 signaling pathway. In vitro, Astragaloside and Atractylenolide (CAA), the main components of HBD, inhibited the proliferation of HCT-116 cells as determined by an MTT assay. Furthermore, CAA notably suppressed the protein expression of IL-6R, STAT3, Survivin, and Cyclin D1 induced by IL-6 in HCT-116 and RAW264.7 cells. These results suggested that HBD exhibits anti-inflammatory and anti-proliferative effects, inhibiting the development of CAC in mice.

## 1. Introduction

Ulcerative colitis (UC) is a chronic non-specific intestinal inflammatory disease with unknown etiology. Lesions of this disease are mainly confined to the mucosa and submucosa of the large intestine, ranging from the distal colon and the proximal segment. The incidences of UC have increased in recent years [1]. Colorectal carcinoma is the most severe prognosis for UC, and the incidence of colitis-associated cancer (CAC) is 4 to 10 times higher than sporadic colorectal cancer. It ranks as the third most common malignant tumor in developed countries, such as European nations and the United States. CAC is a leading cause of death [2]. At present, the treatment of colorectal cancer is mainly based on surgical treatments, such as laparoscopic surgery for primary tumors, and surgical resection is employed for treating metastatic tumors affecting organs, such as the liver and lungs. Radiotherapy is applied for the treatment of rectal cancer and some forms of metastatic disease, such as neoadjuvant and palliative chemotherapy for rectal cancer [3]. Numerous studies have shown that persistent inflammation promotes tumor formation by inducing pre-cancerous cell proliferation and anti-apoptotic properties [4]. Nearly 40% of patients with colitis eventually develop CAC [5]. Chronic intestinal inflammation facilitates pro-inflammatory cytokine production and cell proliferation, and affects immune responses, promoting tumorigenesis in CAC [6,7]. Therefore, therapeutic strategies based on anti-inflammation and mucosa repair in the gut may be effective ways to reduce the incidences of CAC. Signal transducer and activator of transcription (STAT) 3 plays a key role in the host response to injury. The IL-6/STAT3 signaling pathway has also been implicated as a link between inflammation and tumor development. Currently, available strategies include anti-IL-6 or anti-IL-6r antibodies, selective small molecule JAK inhibitors or soluble gp130Fc, a designer cytokine that specifically binds IL-6/6R complexes [8]. In clinics, a traditional Chinese medical prescription has been used for thousands of years to prevent and treat various diseases. It can have a unique and broad spectrum of pharmacological effects by regulating multiple targets [9], and more evidence is needed to prove it. Based on traditional Chinese medicine, spleen deficiency coupled with qi stagnation is one of the most important pathologies of CAC. Treatments for reinforcing qi and/or improving the condition of the spleen are often applied in clinical settings for patients with chronic colitis or colorectal cancer [10]. Huangqi Baizhu decoction (HBD) is a classic prescription originally reported in the book “Su Wen Bing Ji Qi Yi Bao Ming Ji”. It is also a common therapeutic compound for treating colitis and CAC accompanied with spleen deficiency and qi stagnation syndrome. HBD consists of *Radix Astragali* (Huangqi) and *Rhizoma Atractylodis* (Baizhu), which have notable anti-inflammatory properties [11]. *Radix Astragali* has anti-inflammatory and anti-cancer properties. *Rhizoma Atractylodis* regulates immunity and exhibits anti-tumor and anti-oxidative effects. The main component of *Radix Astragali* possessing anti-cancer effects is Astragaloside (AST) [12]. In vitro studies have shown that AST can enhance Cetuximab-mediated inhibition of proliferation and promotes autophagy of the human colon cancer cell line RKO [13]. Atractylenolide II (the main component of *Rhizoma Atractylodis*) has an anti-proliferative effect on digestive tumors [14].

Whether HBD treatment could inhibit the development of CAC and its underlying mechanisms needs to be determined. Thus, in this study, we evaluated the effects of HBD in AOM/DSS-induced CAC mice and HCT-116 cells, and the underlying mechanism was clarified for the first time.

## 2. Results

### 2.1. Huangqi Baizhu Decoction Inhibited the Occurrence and Progression of Colitis-Associated Cancer

In this study, we used an AOM/DSS murine model to investigate the effects of HBD on colitis-associated cancer, and the underlying mechanisms were explored. The experimental protocol of AOM/DSS HBD treatment is summarized in Figure 1A. As shown in Figure 1C, the body weight of AOM/DSS mice was reduced during the experimental eight-week period compared with the control mice. Compared with the AOM/DSS group, the Huangqi Baizhu treatment group showed an increased survival rate throughout the course of experiment, as shown in Figure 1B. The ratio of colon weight/colon length, which is also considered as an indicator of inflammation, was observed in colitis [15]. As shown in Figure 1D,E, the ratio of colon weight/colon length in the AOM/DSS group was significantly greater than that of the normal group. The weight to length ratio of the Huangqi Baizhu low concentration (HBL) group is lower than that of the model group. The deteriorating effects of AOM/DSS treatment on the colon were reversed with low concentration, but not high concentration treatments. As shown in Figure 1E–G, the number, size and location of detectable tumors were examined. The average size of tumors of the AOM/DSS group mice is 0.95 ± 0.28 cm^2^, while that of the HBL and HBH groups is 0.44 ± 0.17 cm^2^ and 0.57 ± 0.21 cm^2^, respectively. Taverage tumor size of the SASP group is 0.37 ± 0.20 cm^2^. The average number of tumors of the AOM/DSS group is 4.86 ± 1.46, and that of the HBL and HBH groups is 2.57 ± 0.79 and 3.2 ± 1.48, respectively, while that of the SASP group is 2.2 ± 0.84. As the data indicate, there were fewer tumors and smaller tumor sizes in mice of the Huangqi Baizhu low concentration (HBL) and SASP groups than the AOM/DSS treatment group.

### 2.2. HBD Inhibits Tumor Proliferation in Colitis-Associated Cancer, and Downregulates the Expression of JAK2 and P-STAT3 in IL-6/STAT3 Pathway

As shown in Figure 2A–E,G, HE staining and the histological scores show that the degree of atypical hyperplasia in the model group was significantly higher than the normal group (*p* < 0.05). The degree of dysplasia in the HBL and SASP groups was lower than the model group. However, the results were not statistically significant (*p* > 0.05). As shown in Figure 2F,H–J, we measured the expression levels of Ki-67, p-STAT3, and JAK2 in the different groups. Immunohistochemistry showed that HBL treatment significantly inhibited cell proliferation compared to the AOM/DSS model group (*p* < 0.05). The expression levels of p-STAT3 and JAK2 in the HBL and SASP groups were also significantly lower than the model group (*p* < 0.05). It was found that HBL and SASP decreased the expression of p-STAT3, Ki-67, and JAK2 compared with the AOM/DSS group. These results indicate that HBL inhibits expression of colitis-associated proteins involved in the IL-6/STAT3 signaling pathway in the CAC model.

### 2.3. HBD Attenuates the Expression of Inflammatory Factors

Pro-inflammatory cytokines are key signaling molecules in intestinal immunity and are known to participate in the disruption of controlled inflammation [16]. TNF-α, IL-1β, IL-6, IL-17, and IL-21, have been detected in the mucosal tissues of patients with CAC [17]. Figure 3A,B, show the relative mRNA expression levels of TNF-α and IL-1β in the control, experimental and Huangqi Baizhu-treatment groups of mice as detected by qRT-PCR analysis. The expression of TNF-α and IL-1β in mice treated with AOM/DSS was significantly increased in comparison with untreated mice. Treatment with HBL and SASP significantly attenuates the expression of TNF-α (*p* < 0.05) and IL-1β (*p* < 0.01), but no significant differences in expression were noted for the Huangqi Baizhu high concentration group (HBH), (*p* > 0.05). However, the mRNA expression levels of IL-6 in the CAC and treatment groups notably changed compared with the normal group (Figure 3C).

### 2.4. HBD Attenuates the IL-6/STAT3/Survivin/Cyclin D1 Signaling Pathway

Our immunohistochemical results showed that HBD reduced the expression of JAK2 and P-STAT3 associated with the IL-6/STAT3 signaling pathway and inhibited the formation of tumor cells in colitis-associated colon cancer. Therefore, the following was determined, first, whether HBD inhibits other related proteins of the IL-6/STA3 signaling pathway, and whether HBD inhibits the expression of genes involved in tumor proliferation. To begin with, we examined the expression levels of STAT3 and phosphorylated STAT3 in the AOM/DSS model and the drug-treated groups by Western blotting. We found that HBD inhibited the expression of P-STAT3 (Tyr-705) and STAT3. Then, we examined the expression levels of GP130 and IL-6Rα, the upstream proteins of STAT3. We analyzed the expression of Cyclin D1 and Survivin, which are involved in tumor proliferation and are downstream molecules of STAT3 signaling. The data showed that HBD treatment significantly decreased the levels of P-STAT3, Cyclin D1, STAT3, GP130, IL-6Rα, and Survivin (Figure 4). The expression of IL-6Rα in the model group was significantly higher than the normal group (*p* < 0.01). The expression of IL-6Rα in the HBD group was lower than the model group (*p* < 0.05).

### 2.5. Atractylenolide II and Astragaloside Inhibited HCT-116 Cell Proliferation

The MTT assay showed that Atractylenolide II and Astragaloside significantly inhibited cell viability in time- and dose-dependent manners. The absorbance was measured at concentrations of 12.5 µg/mL, 25 µg/mL, and 50 µg/mL for 12 h, 24 h, and 48 h. Reduced viability of HCT-116 cells was most obvious after 24 h of treatment with the three drug concentrations. After 48 h post-administration, the inhibitory effects of 25 µg/mL and 50 µg/mL were obvious (Figure 5).

### 2.6. Atractylenolide II and Astragaloside Suppress STAT3 Activation by Inhibiting the Activity of the IL-6Rα/STAT3 Pathway

Dysregulation of the IL-6/STAT3 signaling pathway has an important role in the development of colorectal cancer. Western blot analysis showed that the expression levels of STAT3 in HCT-116 and RAW264.7 cells stimulated by IL-6 increased, and IL-6-induced STAT3 activation was significantly inhibited by ATR II+AST treatment. The expression levels of IL-6Rα, STAT3, Cyclin D1, and Survivin were also increased after stimulation with IL-6. In contrast, ATR II+AST reduced the expression levels of these proteins (Figure 6). In conclusion, ATR II+AST inhibits the activation of the STAT3 signaling pathway induced by IL-6-stimulation of HCT-116 and RAW264.7 cells.

## 3. Discussion

CAC occurs in the rectum and sigmoid colon, the colonic mucosa, submucosa, and muscular layers. CAC can be diffusely distributed within these tissues. According to a large-scale epidemiological survey, the course of disease is the most important factor in the carcinogenesis of ulcerative colitis. The longer the course of disease, the higher the risk of tumorigenesis. Colitis-associated cancer patients are mainly treated by surgery, and are supplemented with other therapeutic strategies, such as chemotherapy and radiotherapy. However, few chemotherapeutic drugs are available and have significant toxic side effects that can severely impair the immune system and hematopoietic system [18,19,20]. Traditional Chinese medicine is effective in the treatment of UC, as reported in clinical settings and experimental research [21,22]. Therefore, many scholars have tried to find drugs that have preventive effects against CAC in traditional Chinese medicine. Some researchers have indicated that HBD has notable inhibitory effects on ulcerative colitis lesions [7]. Total saponins extracted from the medicinal herb *Radix Astragali* exhibits significant growth-inhibitory and proapoptotic effects on human colon cancer cells [23]. Astragaloside IV significantly inhibits the expression of several key cell cycle-related proteins (Cyclin D1 and CDK4) in colorectal cancer cell lines [24]. Paeoniflorin (PF), an active compound in *Rhizoma Atractylodis* macrocephala, inhibits atherosclerosis by suppressing inflammation-related pathways [25], Li N et al. reported that lactones of Atractylodis have notable anti-cancer activity on human cancer cell lines (MCF7, Hep G2, Du145, Colon205, A549, and HL-60) [26]. However, the effects of *Radix Astragali* in combination with *Rhizoma Atractylodis* on treating colitis-associated cancer have not been explored until now.

We observed that the survival rate of mice in the HBD group improved, and the number of tumors, tumor size, and weight/length values of the colon significantly improved compared to the model group with AOM/DSS-induced CAC (Figure 1). In Figure 2A, in mice of the control group, the mucosa of the colon was intact, and no dysplasia in the epithelium or inflammatory cell infiltration was observed. The glands of the colon were uniformly arranged. In the colonic tissues of AOM/DSS-induced mice, most of the epithelia exhibited moderate or severe dysplasia, gland fusion, irregular arrangement and focal chronic inflammatory cell infiltration of the mucosa. Under treatment with AOM/DSS, HBD can significantly reduce the severity of glandular hyperplasia and the degree of inflammatory cell infiltration in the mucosa. In summary, HBD can alleviate the symptoms caused by AOM/DSS in mice. Environmental factors may play key roles in the carcinogenesis of colon tissues. Chronic inflammation is closely related to the progression of this process. The release of inflammatory factors is an important cause of tumorigenesis and tumor development. It activates inflammation-related pathways in an autocrine manner and promotes cell proliferation, leading to the development of tumors. Many studies have shown that downregulating the expression or the knockout of genes encoding carcinogenic inflammatory factors can effectively reduce the occurrence and invasiveness of tumors [27,28]. To confirm our observations, we carried out a series of quantitative analyses. HE staining and immunohistochemical analysis showed that HBD markedly alleviated the extensive infiltration of inflammatory cells and significantly inhibited the expression of Ki-67 induced by DSS (Figure 2A–D). The expression of Ki-67 is closely related to cell proliferation. As cells proliferate, Ki-67 expression also increases, which is inevitable in the progression of the cell cycle [29]. Furthermore, RT-PCR analysis showed that HBD effectively reduced the expression levels of the pro-inflammatory cytokines IL-1β and TNF-α (Figure 3).

The IL-6/STAT3 signaling pathway is a classical inflammatory pathway involved in tumor formation. Therefore, in this study, we examined the effects of HBD on the IL-6/STAT3 signaling pathway under inflammatory conditions in vivo and in vitro. In the process of colitis development, IL-6 binds to soluble or membrane-bound IL-6 receptor (IL-6R) polypeptides that signal by interacting with the membrane-associated gp130 subunit, whose engagement triggers the activation of Janus kinases (JAKs), the downstream effectors of STAT3, Shp2-Ras, and phosphatidylinositol 3-kinase (PI3K)-Akt [30]. After phosphorylation (activated STAT3), and homodimerization or heterodimerization with STAT1 in the cytoplasm, STAT3 translocates to the nucleus to regulate the expression of various genes, including genes that encode apoptosis-related proteins and cell cycle regulators, i.e., Bcl-2, Bcl-xl, mcl-1, Survivin, and Cyclin D1 [31]. Persistent STAT3 target gene activation can stimulate cell growth, angiogenesis, and cell movement, while preventing apoptosis, thereby driving tumorigenesis. The role of STAT3 downstream of IL-6-elicited tumorigenesis has been reported in previous studies [32,33]. Our results confirmed that HBD inhibited the activation of STAT3 and downregulated the expression levels of proteins involved in the IL-6R/STAT3 signaling pathway (Figure 2B,E,F and Figure 4). In vitro, the MTT assay indicated that ATR II +AST significantly inhibited the proliferation of HCT-116 cells, which occurred in a dose- and time-dependent manner (Figure 5). Western blotting showed that ATR II +AST reduced the activation of STAT3 and the expression of IL-6R/STAT3 signaling proteins, such as IL-6Rα, Survivin, and Cyclin D1 (Figure 6).

In vivo, it was found that low and high concentrations of HBD had inhibitory effects on different indicators of CAC, but not all of them. Therefore, we believe that this experiment did not find an optimal concentration for the treatment of CAC and needs further confirmation.

In summary, our results showed that HBD inhibits the carcinogenicity elicited by AOM/DSS, mainly by inhibiting the IL-6Rα/STAT3 signaling pathway.

## 4. Materials and Methods

### 4.1. HBD Preparation

HBD is composed of two traditional Chinese medicines, *Radix Astragali* (huangqi) and *Rhizoma Atractylodis* (baizhu), which was purchased from the Caizhilin chain pharmacy in Guangzhou, China. The components were mixed to a ratio of 3(huangqi):1(baizhu) and soaked in pure water (10 times volume) for half an hour. The water was removed and filtered. The mixture was concentrated to 100 mL in a vacuum rotary evaporator, and dried in a freeze dryer. The prepared lyophilized powder was stored at −20 °C until use. When used, a suspension was made with pure water (0.6 g/mL, HBL; 1.2 g/mL, HBH) for intragastric administration. The dosages of HBL and HBH were 10 and 20 times the adult dose in clinic, respectively. Sulfasalazine enteric-coated tablets (SASP) is a conventional drug for treating UC in clinics. Sulfasalazine enteric-coated tablets (0.03 g/mL, Fuda Pharmaceutical Co.Ltd, ShangHai, China)) were used as a control treatment. The dosages of SASP were 10 times the adult dose in clinics.

### 4.2. Animal Experiment

16–20 g C57BL/6 male mice were obtained from the laboratory animal center in Guangdong, China and fed an adjusted diet for a week in a specific pathogen free (SPF) breeding room of the laboratory animal center at the Guangzhou University of Chinese Medicine. On day 1, 12 mg/kg AOM (Concentration:1 mg/mL) was injected and mice were then maintained on a regular diet with water for 2 days. On day 3, drinking water in cages was replaced with 3% DSS formula for 5 days. After this, mice were given water for 16 days. This was the end of cycle 1 of the treatment program. In the DSS treatment program, the body weight of mice was monitored every day. In the water cycle, mice weight was monitored 3 times per week. If mice lose 10–20% body weight, they should be injected with 1 mL of sterile saline or given wet food. This cycle was repeated twice for a total of three times [34]. All mice of appropriate body weight were stratified and randomly divided into 5 groups (*n* = 20). After 4 weeks, the mice were given 12 g/kg HBD (HBL) and 24 g/kg HBD (HBH). Mice of the CAC and Control groups were administered 20 mL/kg distilled water and 0.6 g/kg sulfasalazine (SASP) intragastrically until analysis (Figure 1A). All animal experiments were approved by the committee of animal ethics of Guangzhou University of Chinese Medicine (NO.2016060, February 29,2016). On day 65, all the mice were fasted for 1 day and then anesthetized by chloral hydrate for sample collection. Detailed information of the medicinal materials and reagents in this study is summarized in Table 1.

### 4.3. Histopathological and Immunohistochemical Analyses

Tissues were immersed in 4% paraformaldehyde and embedded in paraffin wax, sliced with a microtome, stained with hematoxylin and eosin (HE), and observed under a microscope. Colitis-associated cancer pathology was assessed via a 4-point system as follows: 1 = no tumor or dysplasia present; 2 = basally-oriented nuclei, mild nuclear enlargement, nuclear crowding and hyperchromasia, and reduced or loss of intracellular mucin; 3 = prominent nuclear stratification, more severe hyperchromasia and pleomorphism, marked architectural distortion; 4 = back-to-back glands with no intervening stroma, dysplastic epithelial cells, and invasion of the colonic basement membrane [35]. The paraffin-embedded tissue sections were used to detect the protein expression levels of p-STAT3, Ki-67, and JAK2 by immunohistochemistry. After heating at 60 °C in an oven for 30–50 min, tissue sections were placed in xylene and deparaffinized, rehydrated in ethanol/water, and washed with PBS twice (5 min per wash). Then, antigen retrieval solution (Citrate buffer; PH 6.0) was applied to samples for 15 min and boiled, and then taken out at room temperature to cool. Samples were treated with hydrogen peroxide (H_2_O_2_) for 10 min to block endogenous peroxidase activity and the tissue sections were incubated with primary antibodies against Ki-67 (diluted in BSA), JAK2 (diluted in BSA), and p-STAT3 (diluted in BSA) at 4 °C overnight (the concentration is listed in Table 2). The next day, sections were heated at 37 °C in an oven for 30 min. Then, the samples were incubated with a biotin-conjugated secondary antibody and streptavidin-biotin peroxidase, each for 30 min. 3,3′-Diaminobenzidine tetrahydrochloride (0.05%, DAB) was used as the substrate, and a positive signal was detected as a brown color under a light microscope. Counter staining was performed using hematoxylin. Finally, samples were soaked in xylene and ethanol, and photographed with a light microscope. For the immunoreactive score, the scores for the percentage of positive cells and staining intensity were multiplied. Each section was analyzed in five different fields using Image Pro Plus software. The density of yellow reflects the expression levels of target proteins. IOD SUM/Area SUM was applied to quantify the expression of Ki-67, JAK2, and P-STAT3.

### 4.4. Cell Viability Assays

An MTT assay was used to analyze the effects of Atractylenolide II and Astragaloside on the viability the colon cancer cell line HCT-116. Cells were cultured in Dulbecco’s modified Eagles medium (DMEM), supplemented with 10% FBS and antibiotics (penicillin/streptomycin) at 37 °C in a humidified atmosphere containing 5% CO_2_. The number of cells was counted under a standard microscope using a 0.10 mm grid manual counting plate. Cells were seeded at 5 × 10^3^ cells/well in 96-well plates, and 200 µL of DMEM was added. After the cells had attached to culture plates, 12.5 µg/mL, 25 µg/mL, and 50 µg/mL were administered at three concentrations and continued for 12 h, 24 h, and 48 h. At the end of the administration time, we discarded the liquid in the well, added 5 mg/mL of 20 µL MTT reagent and 180 µL of DMEM to each well, which continued to culture for 4 h, after which we discarded the liquid in the well, added 150 µL of DMSO per well, and shook it at room temperature. After shaking for 10 min, the absorbance was measured at 490 nm.

### 4.5. Cell Culture and Conditioned Culture

The two cell lines, HCT-116 and RAW264.7, are well characterized by the ATCC. The two cell lines were cultured in Dulbecco’s modified Eagles medium (Thermo Fisher Scientific, Inc. Waltham, MA, USA) supplemented with 10% FBS and 1% penicillin/streptomycin (Life Technologies, New York, NY, USA). The results of the MTT assay showed that RAW264.7 cells had significantly reduced proliferation after 48 h of treatment. The proliferation of HCT-116 cells was further inhibited after 24 h of administration. Therefore, we stimulated two cells lines with 50 ng/mL of IL-6 and administered the compounds of interest. RAW264.7 cells were cultured for 48 h in DMEM supplemented with IL-6 and three dosing concentrations of ATR II + AST, after which cells were harvested for protein and miRNA extraction. HCT-116 cells in DMEM were supplemented with IL-6 and three concentrations of ATR II + AST. After 24 h of culture, cells were collected, and proteins and miRNAs were collected. All cells were cultured under 5% CO_2_ at 37 °C.

### 4.6. Western Blot Analysis

Western blotting was used to detect the expression levels of STAT3, IL-6R, Cyclin D1, Survivin, and P-STAT3 in colon tissues and treated cells. Briefly, the protein in the colon tissue and cells was extracted using a whole protein extraction kit. Protein samples according to 20 µg, 40 µL specifications were made, proteins were separated by SDS-PAGE gel and transferred to a PVDF membrane. After blocking with 5% skim milk or 5% BSA in TBST for 2 h, the membrane was incubated with primary antibodies overnight at 4 °C (anti-IL-6Rα, diluted in 5% NFDM; anti-STAT3, diluted in 5% NFDM; anti-P-STAT3, diluted in 5% BSA; anti-Survivin, diluted in 5% NFDM; anti-Cyclin D1, diluted in 5% NFDM and anti-GP130, diluted in 5% NFDM; concentrations are listed in Table 3). The next morning, membranes were washed with TBST 3 times (5 min per wash), and secondary antibodies were applied for 1.5 h at room temperature (the concentration of the secondary antibodies were all 1:2000). Membranes were then washed with TBST 3 times. The protein-antibody complexes were detected using a western ECL substrate and quantification was performed using Image software.

### 4.7. RNA Extraction and Quantitative Real Time PCR Analysis

Total RNA from frozen tissues was extracted using the Total RNA Extraction Kit (SLNco, Cinoasia, China). Che concentration and quality of RNA samples were evaluated with a Nanodrop 2000 spectrophotometer (Thermo Fisher Scientific, Inc., Waltham, MA, USA). Reverse transcription was carried out to obtain cDNA using the Master Mix kit (Takara, Shiga, Japan) following standard protocols. The mRNA levels of TNF-α, IL-6, and IL-1β in CAC colon samples and cells were assessed using a Step One Plus real-time PCR system (BIO-RAD, CFX96TM Real-time System, C1000TM Thermal Cycle, USA). Detailed information of the primer sequences employed in this study is summarized in Table 4.

### 4.8. Statistical Analyses

Each experiment was repeated at least three times. All the data were displayed as the mean ± standard deviation (SD) and analyzed with SPSS 17.0. One-way analysis of variance (ANOVA) was used for the comparison between multiple groups. A *t*-test was used for comparison between two groups. A value of *p* < 0.05 was considered as statistically significant.

## 5. Conclusions

Our results showed that HBD inhibits the development of CAC by suppressing the IL-6Rα/STAT3/Survivin/Cyclin D1 signaling pathway, suggesting that HBD may be used in the treatment of CAC.

## Figures and Tables

**Figure 1 ijms-20-01069-f001:**
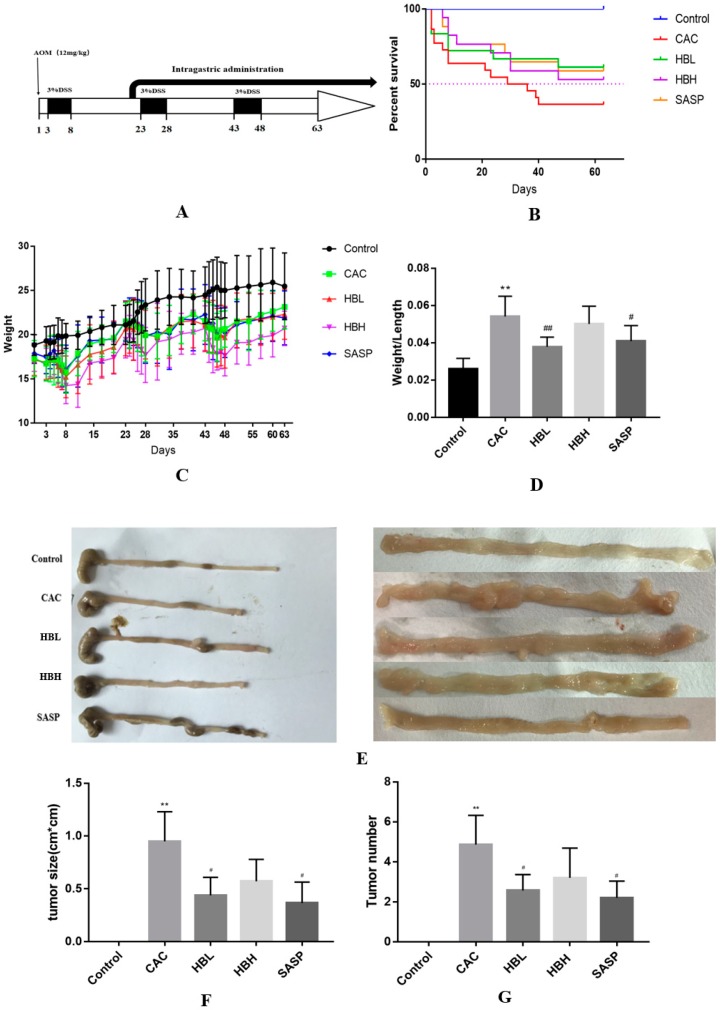
Huangqi Baizhu decoction (HBD) inhibited the occurrence and progression of colitis-associated cancer. (**A**) Experimental protocol for the induction of the colitis-associated cancer (CAC) model in C57BL/6 mice. (**B**) Effects of the model and treatment on the survival of mice. (**C**) Effects of the model and treatment on the body weight of mice. (**D**) The colon weight/length ratio in a mouse model of colitis-associated cancer was significantly increased in animals that received treatment. (**E**) Changes in colon length and colonic morphology. (**F**,**G**) Tumor size and number were significantly reduced in HBL and SASP mice, but was increased in AOM/DSS-induced mice. Data are expressed as the mean ± S.D. Comparisons: ** and * Control vs. CAC, ^#^ CAC vs. HBL, HBH and SASP. ** *p* < 0.01, ^##^
*p* < 0.01, ^#^
*p* < 0.05, * *p* < 0.05. (*n* = 6–8).

**Figure 2 ijms-20-01069-f002:**
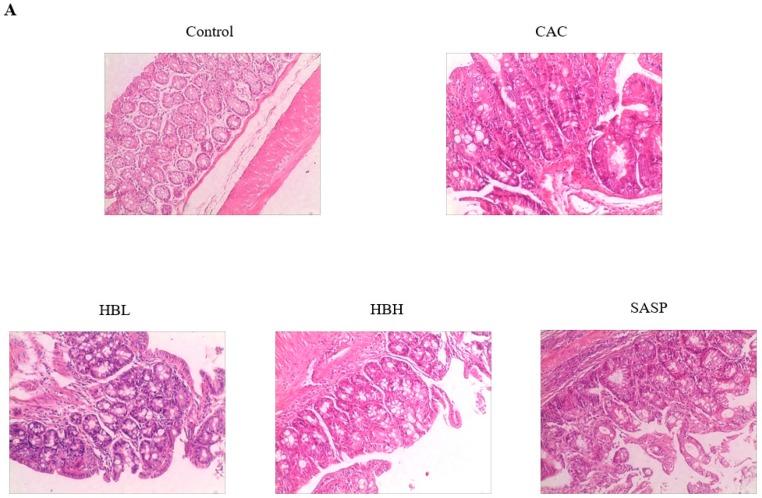
HBD inhibits tumor proliferation in colitis-associated cancer and downregulates the expression of JAK2 and P-STAT3 in the IL-6/STAT3 signaling pathway. (**A**) HE staining (200×) of the colon tissues of the Control, CAC, HBL, HBH, and SASP groups. Location of expression (400×) and levels of Ki-67, JAK2, and P-STAT3 in the colon tissues of the Control, CAC, HBL, HBH and SASP mice as determined by immunohistochemistry (**B**,**D**–**F**). Red arrow indicates the location of positive expression. The data represented IOD SUM/Area SUM. The data of pathological score (**C**). Data are expressed as the mean ± S.D. Comparisons: ** and * Control vs. CAC, ^#^ CAC vs. HBL, HBH and SASP. ** *p* < 0.01, ^#^
*p* < 0.05, * *p* < 0.05. (*n* = 6–8).

**Figure 3 ijms-20-01069-f003:**
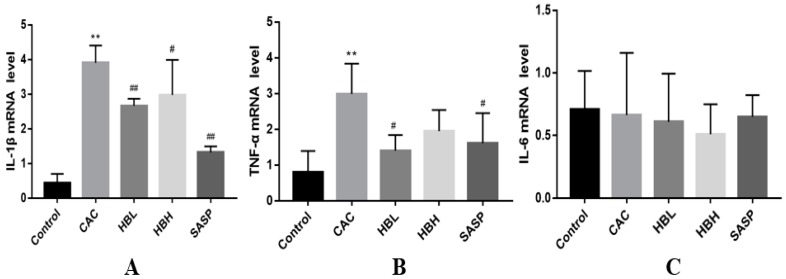
HBD attenuates the expression of inflammatory factors. (**A**–**C**) The mRNA expression levels of inflammatory factors, such as IL-1β, TNF-α, and IL-6, in the colon tissues of mice in different groups. Data are expressed as the mean ± S.D. Comparisons: **Control vs. CAC, ^#^ and ^##^ CAC vs. HBL, HBH and SASP. ** *p* < 0.01, ^##^
*P* < 0.01, ^#^
*p* < 0.05. (*n* = 6–8).

**Figure 4 ijms-20-01069-f004:**
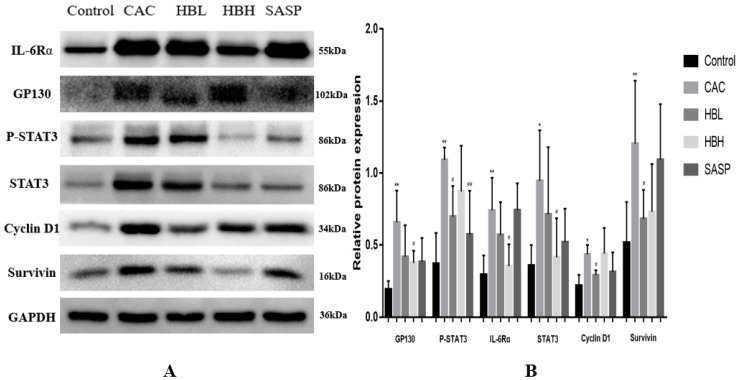
HBD attenuates the IL-6/STAT3/Survivin/Cyclin D1 signaling pathway. Western blot analysis of P-STAT3, STAT3, Survivin, Cyclin D1, GP130, and IL-6Rα in different groups. Data are expressed as the mean ± S.D. Comparisons: ** and * Control vs. CAC, ^##^ and ^#^ CAC vs. Treated group, ** *p* < 0.01, * *p* < 0.05, ^##^
*p* < 0.01, ^#^
*p* < 0.05. (*n* = 6–8).

**Figure 5 ijms-20-01069-f005:**
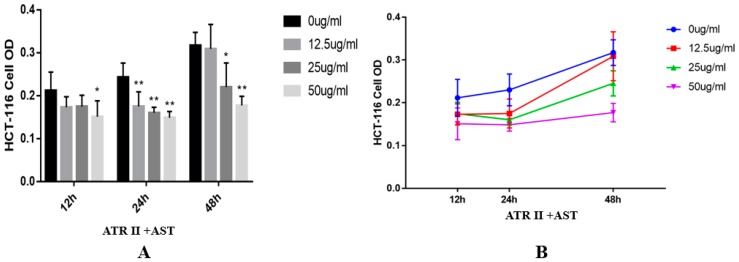
Atractylenolide II and Astragaloside inhibited HCT-116 cell proliferation. (**A**,**B**) Histogram and line chart of the MTT assay results of HCT-116 cells. Data are expressed as the mean ± S.D. Comparisons: ** and * Control vs. Treated group, ** *p* < 0.01, * *p* < 0.05.

**Figure 6 ijms-20-01069-f006:**
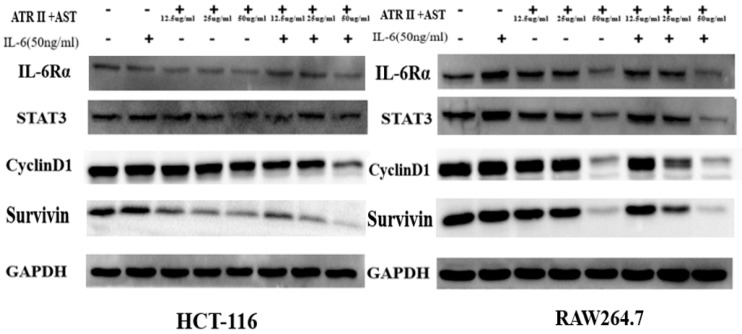
Atractylenolide II and Astragaloside inhibit STAT3 activation by inhibiting the activity of the IL-6Rα/STAT3 signaling pathway. Western blot analysis of STAT3, Survivin, Cyclin D1, and IL-6Rα in HCT-116 and RAW264.7 cells. (*n* = 3–5).

**Table 1 ijms-20-01069-t001:** Medicinal materials and reagents.

Name	Source	
Huangqi	Inner Mongolia	Lot: YPA7E0003
Baizhu	Zhejiang	Lot: YPA7E004
AST	Solarbio	Cat. No.: SA8640
ATR II	Shanghai Yuanye Biological	Lot: PA0808RA13
AOM	Sigma	CAS:25843-45-2
DSS ECL substrate	MP Biomedicals Bio-Rad	Cat. No.: 160110

**Table 2 ijms-20-01069-t002:** Dilutions of primary antibodies for IHC-P.

Antibody	Host	Clonality	Dilution	Source	Cat. No.
Ki-67	Rabbit	Polyclone	1:500	Abcam, USA	Ab15580
Jak2	Rabbit	Monoclonal	1:500	Cell Signaling, USA	D2E12
P-stat3	Rabbit	Monoclonal	1:400	Cell Signaling, USA	D3A7

**Table 3 ijms-20-01069-t003:** Dilutions primary antibodies for WB.

Antibody	Host	Clonality	Dilution	Source	Cat. No.
Survivin	Rabbit	Monoclonal	1:500	Abcam, USA	Ab182132
Survivin	Rabbit	Monoclonal	1:300	Abcam, USA	Ab134170
IL-6Rα	Mouse	Monoclonal	1:500	Santa Cruz, USA	Sc-373708
P-stat3	Mouse	Monoclonal	1:1000	Santa Cruz, USA	Sc-81523
gp130	Rabbit	Polyclone	1:1000	Abacm, USA	Ab202850
Stat3	Mouse	Monoclonal	1:1000	Cell Signaling, USA	124H6
GAPDH	Mouse	Monocloanl	1:5000	Abcam, USA	Ab125247
Cyclin D1	Rabbit	Monoclonal	1:10000	Abcam, USA	Ab134175

**Table 4 ijms-20-01069-t004:** Primer sequence of target genes (All genes species are mouse).

Gene Name	Forward	Reverse
IL-6	CAAGAGACTTCCATCCAGTTGCCT	TTTCTCATTTCCACGATTTCCCAG
TNF-α	CAGGCGGTGCCTATGTCTC	CGATCACCCCGAAGTTCAGTAG
IL-1β	ATGGCAACTGTTCCTGAACTCAACT	AGGACAGGTATAGATTCTTTCCTT
GAPDH	CAAGGCTGTGGGCAAGGTCATCC	TTTCTCCAGGCGGCAGGTCAGAT

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
