# Peer review of "HBD Inhibits the Development of Colitis-Associated Cancer in Mice via the IL-6R/STAT3 Signaling Pathway"

_ijms, 2019, doi:10.3390/ijms20051069_

Round 1

Reviewer 1 Report

The revised manuscript has greatly been improved.

Author Response

Dear reviewer:

             Thanks for you suggestion, based on your suggestion, we have modified the introduction to the article to add the following:

Reviewer 2 Report

This version of manuscript have improved significantly as compared to previous one. Here are few more comments which should be taken care by the authors:

In Materials and Methods, please provide the primers sequences of all the genes used in the manuscript. 

What was the company of ECL substrate?

Seigel and Jemal's 2019 publication is available for cancer statistics. Why 2012 is given in the references, please update. 

Please label the Y-axis of figure 4B.

Please mention about the red arrows in Fig. 2B in Figure labels.

Please correct the spacing problem in manuscript. 

Author Response

Dear reviewer:

     Thanks for you advise, based on you suggestion, we made some modifications. 
